# Towards maximized volumetric capacity via pore-coordinated design for large-volume-change lithium-ion battery anodes

Jiyoung Ma [1], Jaekyung Sung[1], Jaehyung Hong[2], Sujong Chae [1], Namhyung Kim [1], Seong-Hyeon Choi[1], Gyutae Nam[1], Yoonkook Son[3], Sung Youb Kim[2], Minseong Ko[4] & Jaephil Cho[1]

To achieve the urgent requirement for high volumetric energy density in lithium-ion batteries, alloy-based anodes have been spotlighted as next-generation alternatives. Nonetheless, for the veritable accomplishment with regards to high-energy demand, alloy-based anodes must be evaluated considering several crucial factors that determine volumetric capacity. In particular, the electrode swelling upon cycling must be contemplated if these anodes are to replace conventional graphite anodes in terms of volumetric capacity. Herein, we propose macropore-coordinated graphite-silicon composite by incorporating simulation and mathematical calculation of numerical values from experimental data. This unique structure exhibits minimized electrode swelling comparable to conventional graphite under industrial electrode fabrication conditions. Consequently, this hybrid anode, even with high specific capacity (527 mAh g$^{-1}$) and initial coulombic efficiency (93%) in half-cell, achieves higher volumetric capacity (493.9 mAh cm$^{-3}$) and energy density (1825.7 Wh L$^{-1}$) than conventional graphite (361.4 mAh cm$^{-3}$ and 1376.3 Wh L$^{-1}$) after 100 cycles in the full-cell configuration.

[1] Department of Energy Engineering, School of Energy and Chemical Engineering, Ulsan National Institute of Science and Technology (UNIST), 50, UNIST-gil, Ulsan 44919, Republic of Korea. [2] School of Mechanical, Aerospace and Nuclear Engineering, Ulsan National Institute of Science and Technology (UNIST), 50, UNIST-gil, Ulsan 44919, Republic of Korea. [3] Department of Electric Engineering, Chosun University, 309, Pilmun-daero, Dong-gu, Gwangju 61452, Republic of Korea. [4] Department of Metallurgical Engineering, Pukyong National University, Busan 48547, Republic of Korea. These authors contributed equally: Jiyoung Ma, Jaekyung Sung. Correspondence and requests for materials should be addressed to S.Y.K. (email: sykim@unist.ac.kr) or to M.K. (email: msko876@pknu.ac.kr) or to J.C. (email: jpcho@unist.ac.kr)

The ever-increasing demand for high-energy density in lithium-ion batteries has stimulated ongoing research on anode materials. To satisfy this demand, improved anode volumetric capacity in high areal mass loading is a prerequisite for practical full-cell systems[1–4]. However, most studies have focused on increasing the gravimetric capacity, which is the just one of five volumetric capacity determining factors (VDFs) with the active material ratio, initial electrode density, electrode swelling ratio, and N/P ratio (defined by areal capacity ratio between anode and cathode), as shown in Fig. 1a and Eq. (1) (Supplementary Note 1). More critically, although the electrode swelling drastically decreases the volumetric capacity during cycling (Fig. 1b), this value has often been calculated with respect to the volume of the pristine electrode before cycling. Therefore, it is strongly desired that the VDFs-defined volumetric capacity during cycles should be considered.

$$\text{Volumetric capacity} =$$
$$\frac{\text{Gravimetric capacity} \times \text{Active material ratio} \times \text{Initial electrode density}}{\text{Electrode swelling ratio} \times \text{N/P ratio}}$$

$$(1)$$

With regard to volumetric capacity, carbonaceous anode is unprecedented, having low electrode swelling, excellent cyclability, and outstanding densification properties during electrode fabrication[1,5–7]. However, the intrinsic hurdle of low volumetric capacity resulting from limited gravimetric capacity encourages the battery community to focus on alloy-based anodes (Si, Ge, Sn, etc.)[8,9]. Despite their high gravimetric capacity, the most critical challenge concerning alloy-based anodes is poor cyclability due to their massive volume changes during charge–discharge, which induce mechanical fracturing and solid–electrolyte interphase (SEI) layer instability[10,11].

In the past decades, there have been various advanced nano-engineering strategies to alleviate the stress derived from the volume changes, through nanoparticle, nanotube, and nanowire design[12–15]. In particular, yolk–shell nano-structures have been successful, having empty space to accommodate their volume changes, yielding further improved cyclability with high gravimetric capacity[16,17]. However, considering the VDFs, the fabricated electrode yields low volumetric capacity, because nano-properties accompanying the low tap density induce excessive use of binder and conductive agent (low active material ratio) and incompatibility with electrode calendering (low electrode density)[18,19].

Recently, graphite-alloy composites have received considerable attention, because the nano-engineering advantages are strengthened while the nano-property weaknesses are compensated[20,21]. Most previous studies indicate that these composites are the most feasible alternatives for next-generation anodes, having high gravimetric capacity, superior cyclability, and high tap density[22–31]. And previously reported graphite-alloy composites apparently have satisfactory values for several VDFs to yield high volumetric capacity. However, to replace the conventional graphite (G) anode entirely with regard to volumetric capacity, the graphite-alloy composite electrode swelling upon battery cycling should be rectified to become comparable to that of a conventional G electrode under industrial electrode conditions (high areal capacity loading $\geq 3.5$ mAh cm$^{-2}$, high electrode density of 1.6 g cm$^{-3}$, and limited binder, i.e., $\leq 3$ wt%). Thus, there remains strong motivation to develop rationally designed model to negate the electrode swelling of graphite-alloy composite for high volumetric capacity.

In our previous in situ transmission electron microscopy (TEM) analysis, we noted that the Si-layer volume expansion within the mesopore pushes the Si toward the graphite during lithiation[23]. Such phenomenon could directly influence the electrode swelling and electrical contact loss, and damage the graphite, re-exposing the fresh graphite surface to the electrolyte and causing severe volumetric capacity fading during cycling. On the other hand, mathematical calculation confirmed that a macropore can have sufficient space to accommodate expansion of the Si-layers ($\geq 7$ nm) contained within this pore (Supplementary Fig. 1 and Supplementary Note 2). Accordingly, we thoughtfully considered the effective exploitation of pores to negate the electrode swelling from Si within the graphite-Si composite (Supplementary Fig. 3).

Notice that inner pore distribution of G can be manipulated during pyrolysis treatment of carbon precursors. In this study, we design macropore-coordinated graphite (MG) via chemical vapor deposition process which is mature technology currently used in the anode industry. And, macropore-coordinated graphite-Si (MGS) hybrid is developed based on thorough analytical methods and mathematical calculation (Fig. 1c, d and Supplementary Note 3). This unique structure, with no Si-layers located on the mesopores because of carbon pre-filling (carbon-blocking), maintains morphological integrity without cracking and contact losses during cycling. Notably, volume expansion of Si-layers selectively situated on internal macropores is well accommodated. Therefore, the MGS demonstrates a minimized electrode swelling ratio comparable to that of conventional G under industrial electrode fabrication conditions for 500 h (over 100 cycles) according to in situ thickness measurement, as shown in Fig. 1e, f and Supplementary Fig. 4. As a result, the strategically synthesized hybrid anode achieved higher volumetric capacity (493.9 mAh cm$^{-3}$) and energy density (1825.7 Wh L$^{-1}$) than that of G (361.4 mAh cm$^{-3}$ and 1376.3 Wh L$^{-1}$) at 100 cycles in full-cell configuration.

## Results

**MGS fabrication and characterization.** Cross-sectional scanning electron microscope (SEM) images show that particles of spherical-type conventional G, selected as pristine materials for MGS synthesis, contain many internal mesopores and macropores (Fig. 2a). First, the mesopores were filled via carbon-blocking to obtain MG through a chemical vapor deposition process using ethylene ($C_2H_4$), in accordance with the mathematical calculation in Supplementary Note 3 (Fig. 2b). The carbon-blocking prevented subsequent Si coating from filling the mesopores. Si-layers were then homogenously distributed in the macropores and on the G surface via thermal decomposition of monosilane ($SiH_4$) (Fig. 2c). The unique structural characteristics of MGS are schematically illustrated in the cross-section of Fig. 2d. High-resolution TEM (HR-TEM) images (Fig. 2e) and fast Fourier transform analysis show that carbon-blocking with ~20-nm thickness occurred between the G. As shown in Fig. 2f, the presence of carbon-layers (~30 nm) on the external G surface implies that most mesopores (~50 nm) could be filled by carbon-blocking, because the carbon-layers formed on both G sides.

To further verify that the carbon-blocking suitably filled the mesopores, the volume of pores with a diameter of 2–50 nm was investigated using the Barrett–Joyner–Halenda (BJH) method (Fig. 2g). Both the carbon and Si-layer coating on the G, required for the MG and Si-layer-coated G (GS) synthesis processes, respectively, reduced the G pore volume by a similar amount (Supplementary Table 2). These results confirm that the mesopores could be pre-filled through carbon-blocking before Si-layer deposition on the MG, as designed (Supplementary Note 3). However, we observed that the Si-layer coating process after pre-filling with carbon-blocking (for the MGS synthetic process) slightly reduced the pore volume in the mesopore range.

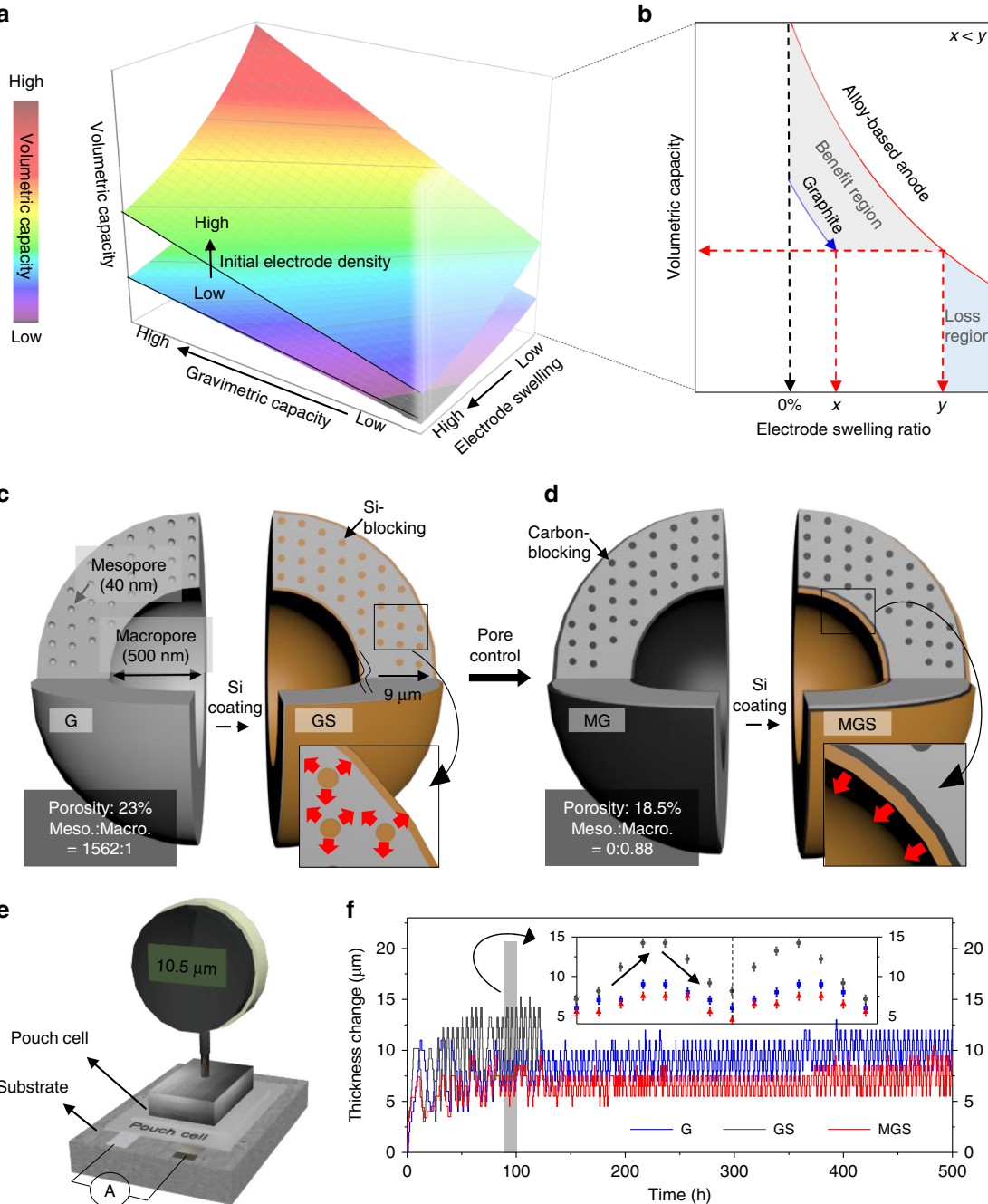

**Fig. 1** Critical factors influencing volumetric capacity and in situ thickness changes for MGS structural design. **a** Three-dimensional (3D) graph, showing a correlation between volumetric capacity and other factors, including gravimetric capacity, electrode swelling ratio, and electrode density. **b** Volumetric capacity curves versus swelling ratio for graphite and alloy-based anode when electrode density is fixed. The benefit (gray) region indicates that the graphite-Si with specific electrode swelling (below $y$) has a higher volumetric capacity than that of graphite with value $x$. **c**, **d** Schematic model systemically computed based on the experimentally analyzed results GS (**c**) and MGS (**d**). (The design description is given in Supplementary Fig. 2, Note 3, and Table 1.) **e** Diagram of in situ thickness measurement system for pouch-type full-cell (G, GS, and MGS) upon cycling. **f** Electrode thickness change times of G, GS, and MGS during 100 cycles. Inset: Two cycles (black arrows indicate thickness changes during charging and discharging) with error bars (±0.5 μm)

This phenomenon was also noted when artificial graphite (aG) without internal pores was coated on the Si-layers (aGS); thus, the reduced pore volume stems from the surface, not the mesopore inside G (Supplementary Figs. 6 and 7).

We performed mercury porosimetry to demonstrate the Si-layer coating on the macropores. Figure 2h indicates that the MG pore volume decreased dramatically following Si-layer coating, implying that most Si-layers filled the macropores. As the overall Si content of MGS is similar to that of GS (Supplementary

Note 3), the majority of the Si located in the mesopores was successfully moved to the macropores by manipulating the pore distribution using carbon-blocking. Therefore, Si-layer pushing of the G within the mesopores (Si-blocking) is completely prevented in MGS. Further, MGS exhibits high tap density, low porosity, and low specific surface area because of the carbon-blocking (Fig. 2i and Supplementary Table 2). These outstanding powder characteristics (low porosity and specific surface area) induce side reaction reduction during electrochemical reactions, yielding a

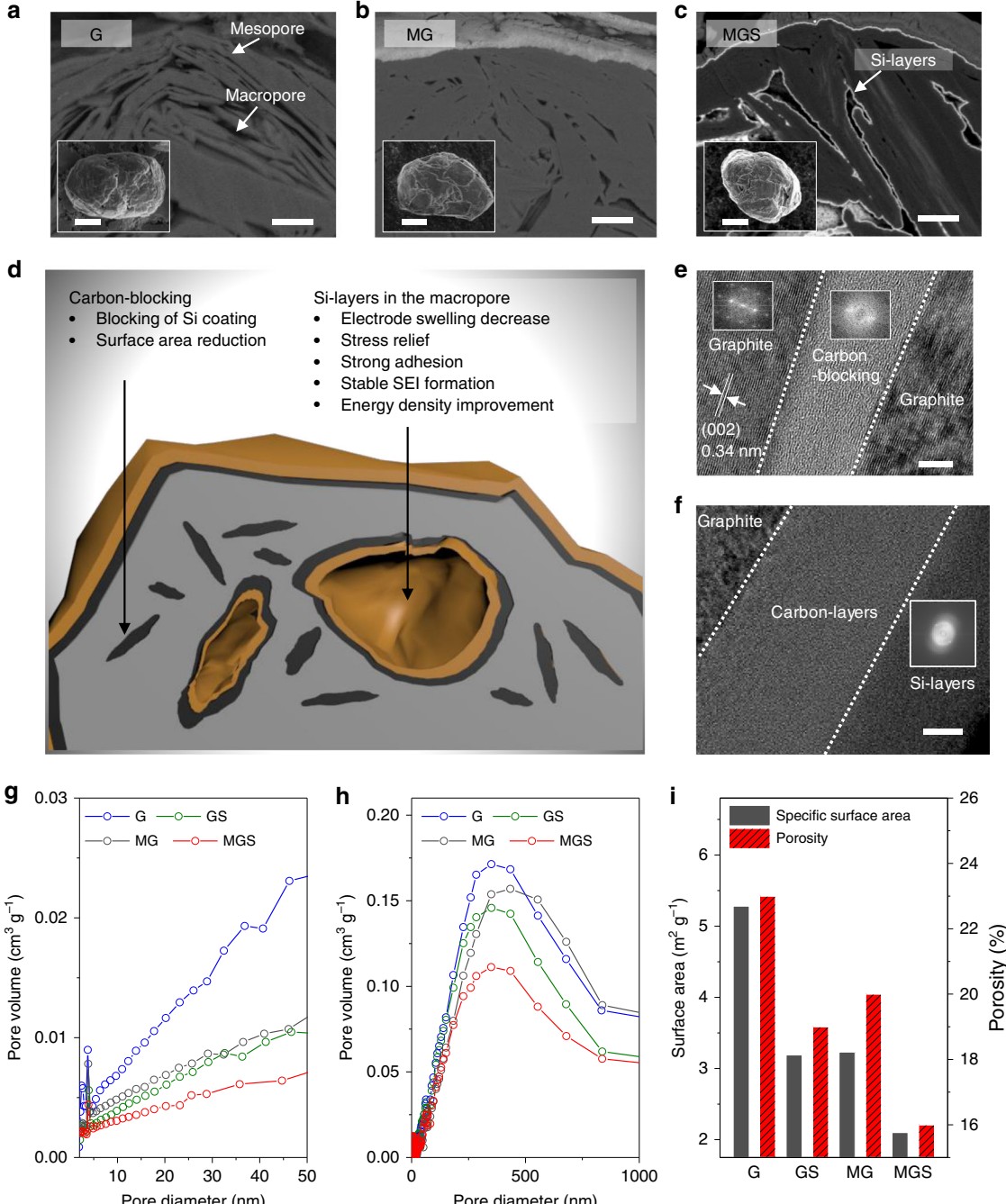

**Fig. 2** MGS fabrication and characterization. Cross-sectional SEM images of G (**a**), MG (**b**), and MGS (**c**), after ion beam milling, with insets showing top views of SEM images. **d** Cross-sectional schematic illustration of MGS providing detailed characteristics of each component. **e, f** High-resolution TEM images at inner region (**e**) and surface (**f**) of MGS with fast Fourier transform inset images. The white line indicates the (002) plane of graphite spanning an inter-layer distance of 0.34 nm. The white dotted lines indicate the boundaries between the graphite, carbon-blocking, and Si. Mesopore size distribution determined via BJH method (**g**) and macropore size distribution obtained through mercury intrusion porosimetry (**h**) of G (black), GS (green), MG (blue), and MGS (red). **i** Specific surface area (black) and porosity (red) of G, GS, MG, and MGS. Scale bars, 1 μm (**a–c**), 5 μm (inset in **a–c**), 5 nm (**e**), 10 nm (**f**)

high initial Coulombic efficiency (CE) of 93% and stable cyclability[32,33].

**Electrode swelling testing and analysis via simulation.** The electrode (G, GS, and MGS anodes) thickness variation in full-cell configuration was examined via SEM before and after 100 cycles in the fully lithiated state (detailed electrode information and electrochemical conditions are given in the Methods section). As shown in Fig. 3a–c, the initial G thickness (63 μm) exceeded that of GS (43 μm) and MGS (43 μm) under industrial electrode conditions. After 100 cycles, the GS-containing electrode revealed a 35% (15 μm) swelling ratio, approximately twice the 17% (11 μm) value for the G electrode (Fig. 3d, e). However, the MGS electrode exhibited only 19% (8 μm) swelling after 100 cycles (Fig. 3f).

To investigate the influence of lithiated Si on the electrode swelling, we calculated the Si distribution of each sample in

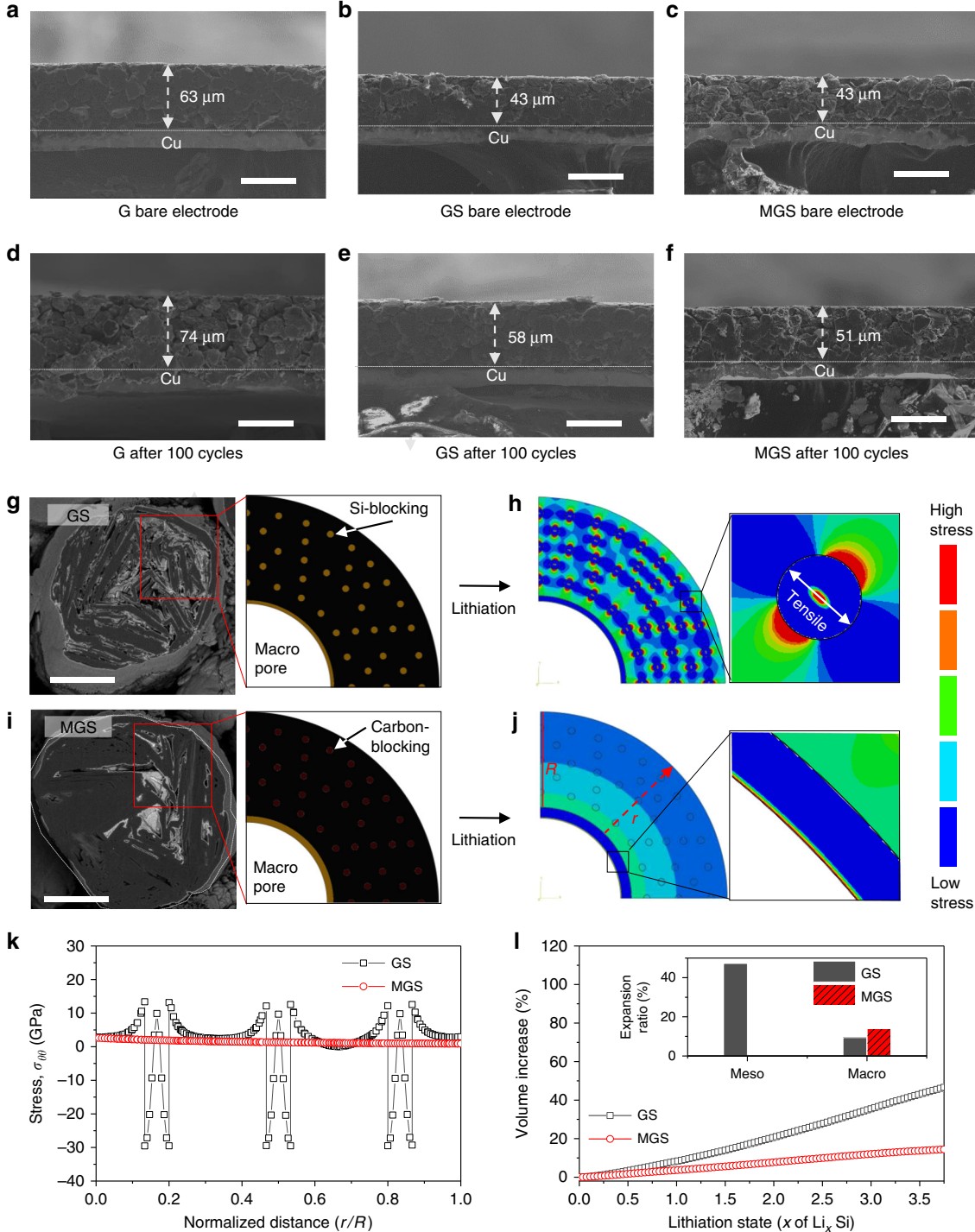

**Fig. 3** Series of cross-sectional SEM images of electrode thickness differences before and after 100 cycles with FEM of GS and MGS upon lithiation. **a–f** Cross-sectional views of G, GS, and MGS electrodes before cycling (**a–c**) and after 100 cycles (**d–f**), respectively. Finite element model for GS (**g**) and MGS (**i**), where photographs show each particle. Diffusion-induced hoop stress in lithiation state (**h**, **j**). The red dotted line indicates the diagonal path. **k** Comparison of hoop stress along with diagonal path. **l** Representative volume increase versus lithiation state. Inset: Expansion ratio depending on different pore types. Scale bars, 50 μm (**a–f**) and 5 μm (**g**, **i**)

accordance with the numerical values from the experimentally analyzed results, e.g., those for the pore volume by BJH, porosity, bulk density, true density and pure Si capacity (Supplementary Note 3). For the GS, Si (total weight fraction in GS is 6.3 wt% indicating 100 vol% as total Si volume) was distributed in the mesopores (2.6 wt%, 40.5 vol%), macropores (2.9 wt%, 46.6 vol%), and on the surface (0.8 wt%, 12.9 vol%). We believe that the Si-blocking in the mesopores strongly contributed to the electrode swelling, as it pushed directly toward the G surface. However, no Si-blocking occurred in the MGS mesopores; the MGS contained Si (the same amount of Si in GS) in the macropores (4.8 wt%, 76 vol%) and on the surface (1.5 wt%, 24 vol%). Note that a large amount of Si in the macropores yields low electrode swelling, because the Si volume expansion is highly mitigated by the

macropore vacant space during lithiation. Nevertheless, the reason why MGS electrode swelling ratio slightly exceeds than that of G is that the one Si-layer on the external surface is likely to influence the electrode swelling (Supplementary Fig. 4). For more details, we conducted numerical simulations using the finite element method (FEM) at single-particle level to further elucidate the electrode swelling (detailed description of numerical simulations is in the Methods section).

Schematic illustrations of pristine GS and MGS particle and cross-sectional SEM images are shown in Fig. 3g, i. Computation of the diffusion-induced stress based on two-phase lithiation of Si[34] with concentration-dependent properties[35,36] indicates that tensile stress (red region) was strongly applied to the GS at the core and the Si-blocking boundary in the fully lithiated ($x = 3.75$, Li$_x$Si) state (Fig. 3h, k and Supplementary Movie 1). In contrast, there was little tensile stress in the lithiated MGS particle (Fig. 3j, k and Supplementary Movie 2). The corresponding volume increase was also calculated through numerical simulation. As shown in Fig. 3l, MGS exhibited a low volume increase ratio arising from stress relaxation due to the macropore vacant space. However, the GS volume increase ratio far exceeded that of MGS because of the high Si-blocking volume expansion in the mesopores. Therefore, stress relaxation due to the accommodation of the Si volume expansion by the macropore empty space induces a low electrode swelling ratio (additional TEM images are given in Supplementary Fig. 17).

**Morphological changes and SEI-layer formation**. To allow comprehensive understanding of both the Si-blocking in the mesopores (GS) and the Si-layers in the macropores (MGS) during cycling, the morphological changes and SEI-layer formation are schematically shown in Fig. 4a, b. Briefly, when the Si-blocking expands towards the G upon lithiation, it applies extremely high tensile stress on the G, inducing crack formation. The stresses allow the Si-blocking to attach strongly to the bottom and upper side of the G during lithiation. The adhesive Si maintains this contact upon contraction during delithiation, causing Si fracturing. After many cycles, the G crack formation and Si-blocking fracturing induces electrical contact loss and newly exposes a fresh surface to the electrolyte, causing continuous SEI-layer formation. Previous studies have reported Si nanoparticle critical sizes of approximately 150 and 300 nm, and Si fracturing can be avoided under these sizes[37,38]. Although <50 nm is far smaller than the Si critical size, isolated Si-blocking may cause morphological deformation and fracture upon cycling (detailed interpretation about the reason of morphological deformation and fracture of Si below 50 nm are given in Fig. 4k–m). On the other hand, the Si-layers in the macropore, having sufficient space to accommodate the volume expansion, generate far less tensile stress on the G upon lithiation. Therefore, no G crack formation and Si-layer fracturing occur during cycling, yielding a relatively stable SEI-layer on the Si and good Si-to-G electrical contact.

Ex situ TEM and SEM images were obtained before and after 100 cycles (delithiated state) in the full-cell configuration, to confirm both the morphological changes and SEI-layer. As shown in Fig. 4c, d, Si-blocking with various thicknesses (≤40 nm) were found inside the GS particle. Si-layers with ~20-nm thickness were on the surface; thus, Si-blocking (~40 nm) could be formed in the mesopores because the Si-layers were coated on both G sides. Further, Si-layers with ~30-nm size were observed inside the MGS macropores (Fig. 4g, h, Supplementary Figs. 8 and 18). As the Si had the same weight percent in the MGS and GS, the MGS Si-layers were thicker than those of the GS because of the intensively coated Si-layers in the macropores (Supplementary

Figs. 9, 13 and 16). After 100 cycles, some G cracks and Si-blocking fracturing were observed, as shown in the SEM images of the GS electrode (Fig. 4e). Furthermore, the TEM images confirm severe Si-blocking fracturing, and related elemental mapping images show fluorine and oxygen elements, which are evidence of SEI-layer[39–41] (Fig. 4f). Since additional SEI layer can occur at the newly exposed, the G cracks and Si-blocking fracturing may induce a continuous SEI-layer during cycling, leading to the low CE and severe capacity fading. On the other hand, the Si-layers in the MGS macropores preserved its morphological integrity without physical damage to the G (Fig. 4i). Therefore, a relatively stable SEI-layer was preserved on the Si-layers, which maintained attachment to the G (Fig. 4j), ensuring high CE and cycle stability (see in situ TEM analysis of Si volume changes in mesopore and macropore during lithiation in ref. [23]).

For detailed understanding of the Si-blocking behavior in the mesopores, we simulated the lithiation/delithiation process of the mesopore Si-blocking through FEM. Noting that the critical fracture energies for Si and G crack formation are reported to be 9 and 17–69 J m$^{-2}$, respectively[42,43], we examined the possible growth of a crack applied to the Si and G adjacent to the Si-blocking. The calculated Si and G fracture energies with a pre-existing 5 nm-sized crack in GS during lithiation were 10.44 and 71.04 J m$^{-2}$, respectively (6.88 and 3.80 J m$^{-2}$, respectively, in MGS); thus, results exceeding the critical values indicate crack growth during the lithiation/delithiation process. Furthermore, compressive stress (blue region) was strongly applied to the G around the Si-blocking in the lithiated state, suggesting that the Si was strongly attached to the bottom and upper G, as this region functions as a cohesion zone (Fig. 4k). For delithiation, the stress distribution was opposite to the lithiation case. Consequently, tensile stress, which functions as a crack expander in delithiation, was applied to both Si-blocking sides (Fig. 4l, m). These results coincide with the Fig. 4a illustration.

**Electrochemical characterization considering electrode swelling**. To confirm MGS anode practical viability, electrochemical tests were performed in half- and full-cell configuration under industrial electrode conditions (see the additional data in Supplementary Figs. 11, 12 and Supplementary Note 4). The reversible capacities of the synthesized MGS, GS, and conventional G were measured using the constant current and constant voltage technique, from 0.005 to 1.5 V (first cycle in half-cell). The MGS, GS, and G exhibited first-cycle gravimetric reversible capacities of 527, 525, and 360 mAh g$^{-1}$, respectively, with initial CEs of 93.0%, 92.2%, and 92.0%, respectively (Fig. 5a). Based on the half-cell electrochemical results, we designed a pouch-type full-cell containing MGS anode and LiCoO$_2$ (LCO) cathode (Supplementary Fig. 10). The anode and cathode areal charge capacities were fixed at 3.8 and 3.5 mAh cm$^{-2}$, respectively, yielding a theoretical N/P ratio of 1.1. As shown by the normalized capacity profiles (Fig. 5b), MGS revealed an initial CE of 91.3% with a 3.16 mAh cm$^{-2}$ discharge capacity, exceeding that of G (3.11 mAh cm$^{-2}$ with 90.0%) for the first cycle within a voltage range of 2.5–4.35 V (detailed results of full-cell comprising MGS and LiNi0.6Co0.2Mn0.2O$_2$ are given in Supplementary Fig. 14). The low specific surface area, which reduced the side reaction during electrochemical reaction, caused the increased initial CE of MGS. The volumetric capacity was defined as the discharge capacity divided by the maximum thickness during cycling, which was determined through in situ thickness measurement for 100 cycles.

As shown in Fig. 5c, MGS exhibited a higher initial volumetric capacity (632.0 mAh cm$^{-3}$) than GS (588.6 mAh cm$^{-3}$). Even after 100 cycles, the MGS volumetric capacity (493.9 mAh cm$^{-3}$)

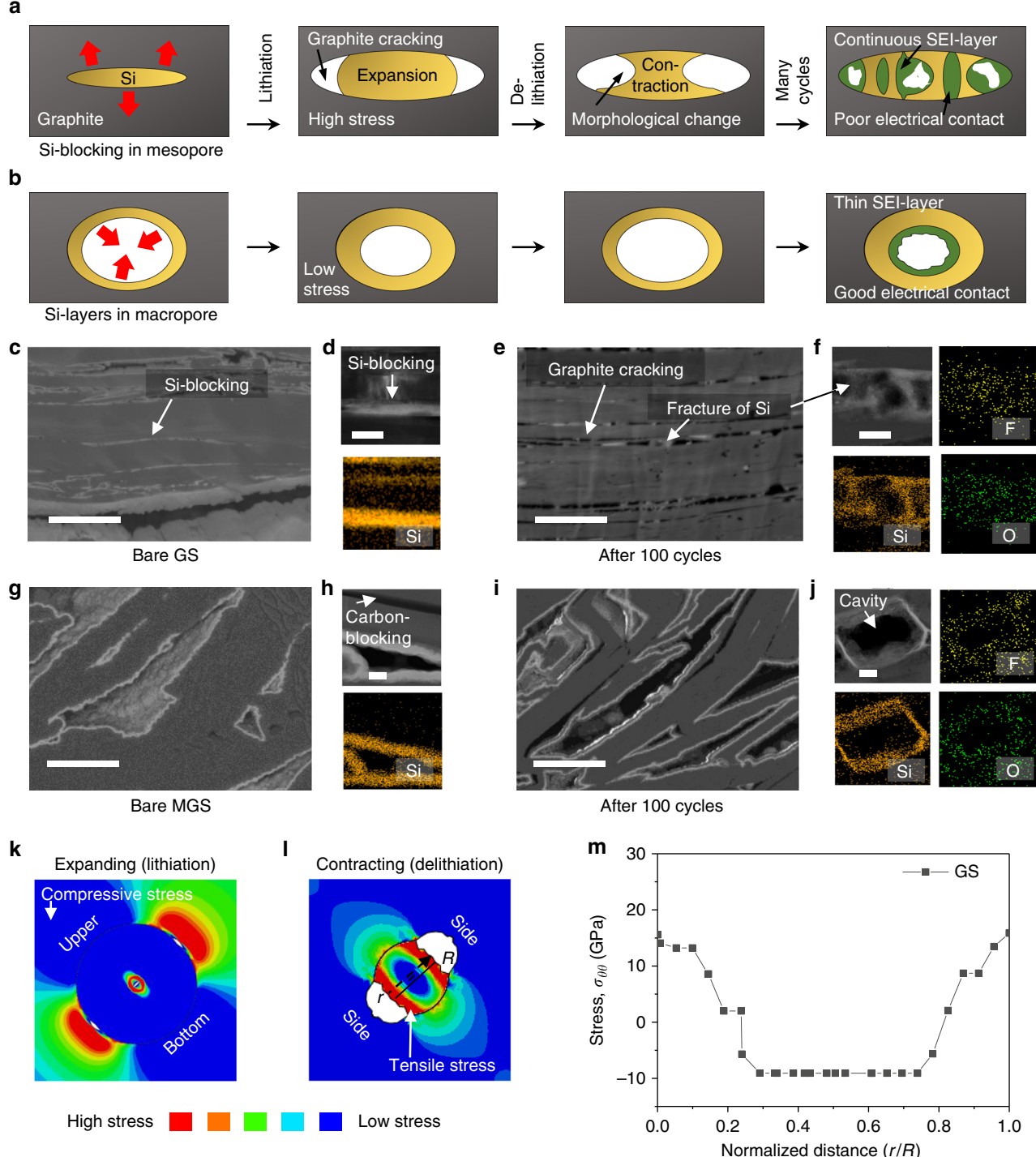

**Fig. 4** Structural changes and SEI formation of GS and MGS before and after 100 cycles with FEM of GS upon delithiation. **a, b** Schematic of SEI formation of Si in mesopore (**a**) and macropore (**b**). **c–j** Cross-sectional SEM and scanning TEM (STEM) images after ion beam milling with elemental mapping through energy-dispersive spectroscopy of GS and MGS electrode before cycling (**c, d, g, h**) and after 100 cycles (**e, f, i, j**), respectively. **k, l** Estimation of structural change based on finite element results during lithiation and delithiation. **m** FEM of diffusion-induced hoop stress along with diagonal path in GS delithiation state. Scale bars, 1 μm (**c, e, g, i**) and 50 nm (**d, f, h, j**)

exceeded that of GS (367.5 mAh cm$^{-3}$), because of the low swelling ratio with improved cyclic stability (Figs. 1f and 5e, Supplementary Fig. 15). Moreover, MGS exhibited a remarkable CE increase exceeding 99.5% after only 4 cycles, whereas the CE of GS was only reached after 10 cycles. These results imply that the macropores relieve the stress due to Si volume expansion, allowing the formation of a stable SEI-layer during cycling.

Although the GS seemed to exhibit higher volumetric capacity (588.6 mAh cm$^{-3}$) than G (420.2 mAh cm$^{-3}$) before cycling, the GS value (367.5 mAh cm$^{-3}$) became similar to that of G (361.4 mAh cm$^{-3}$) after 100 cycles due to high electrode volume expansion with poor cycle retention. These results imply that volumetric capacity should be considered based on electrode swelling and capacity retention after cycling. As shown in Fig. 5d,

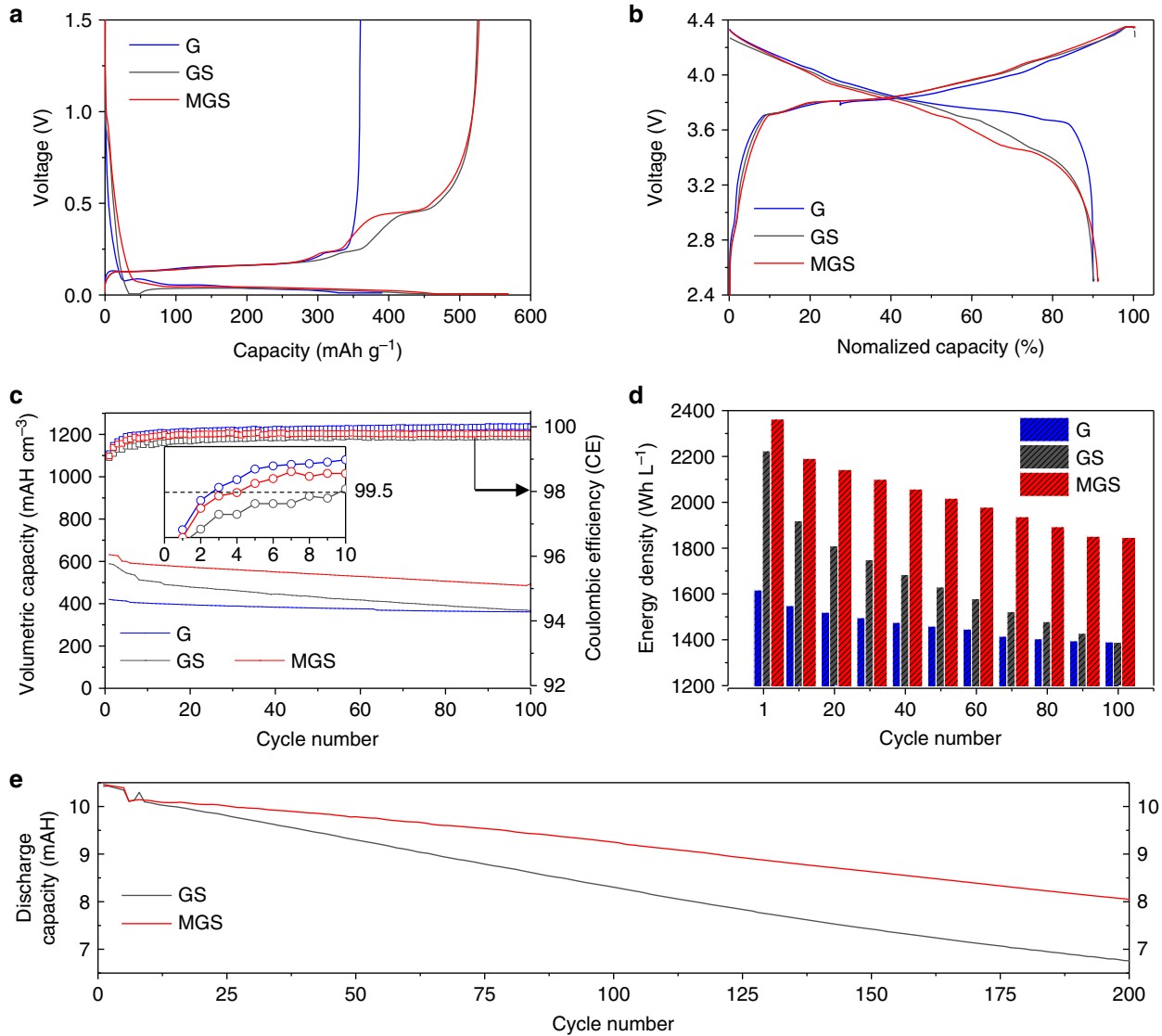

**Fig. 5** Electrochemical characterization of various anodes. Voltage profiles of G, GS, and MGS half-cell (**a**) and full-cell (**b**) plotted for first cycles at 0.1 C. **c** Full-cell volumetric capacity as reversible areal capacity divided by maximum electrode thickness at each cycle for 100th cycle at 1 C discharge rate and 0.5 C charge rate in 2.7–4.35 V potential range. The mass loading of the GS and MGS was 6.9 mg cm$^{-2}$ (G: 10.1 mg cm$^{-2}$). The CE of each sample is plotted on the secondary y-axis. Inset: Magnified graphs of CE with 99.5 line for 10 cycles to indicate the number of cycles for CE to reach 99.5%. **d** Volumetric energy densities for 1st, 20th, 40th, 60th, 80th, and 100th cycles. **e** Full-cell discharge capacity of GS and MGS paired with conventional lithium cobalt oxide. The mass loading of the GS and MGS was 3 mg cm$^{-2}$

we proved that MGS has an outstanding energy density of 1825.7 Wh L$^{-1}$ being higher than those of G and GS (1376.3 and 1374.5 Wh L$^{-1}$, respectively). Detailed electrode and energy density information is given in Supplementary Table 3.

## Discussion
We introduced strategically designed MGS through pore-distribution manipulation, based on a calculation model from experimentally analyzed results with insight into the vacant space in composite. Interestingly, the developed MGS exhibited a minimized electrode swelling ratio (19%) comparable to that of conventional G (17%) even after 100 cycles in full-cell configuration under industrial electrode fabrication conditions (≥ 3.1 mAh cm$^{-2}$, 1.6 g cm$^{-3}$). In addition to the high initial CE of 93.0% and 527 mAh g$^{-1}$, a rapid cycling efficiency increase exceeding 99.5% over 4 cycles was exhibited, because of the crack- and contact-loss-free morphological integrity during cycling.

These outstanding features eventually demonstrated a volumetric capacity of 493.9 mAh cm$^{-3}$ and energy density of 1825.7 Wh L$^{-1}$ exceeding those of G (361.4 mAh cm$^{-3}$ and 1376.3 Wh L$^{-1}$) after 100 cycles. Further, our material design satisfies the most rigorous requirements concerning high-energy density, being sufficiently large to replace a conventional G anode even after cycling. Thus, the exquisitely designed MGS paves the path for next-generation anodes with both high-energy density and commercial feasibility; this could also be a significant breakthrough for the development of an electric vehicle with more extended driving mileage of several times than before on a single charge.

## Methods
**Synthesis of MGS.** For the fabrication of MGS, 5 kg of spherical graphite was introduced into a rotary tube furnace. To fill mesopore with carbon-blocking, thermal decomposition of ethylene gas (99.9%) was executed at 900 °C for 3 h (5 L min$^{-1}$). In succession, high-purity monosilane gas (99.9999%) flowed in the same furnace at 475 °C for 1 h (5 L min$^{-1}$). In case of all samples for electrochemical test, additional carbon coating was performed for 2 h (5 L min$^{-1}$).

**Material characterization**. The samples were visualized using scanning electron microscopy (Verios 460, FEI) with energy-dispersive spectroscopy (XFlash 6130, Bruker) and high-resolution transmission electron microscopy (JEM-2100F, JEOL). Sample preparation for the cross-sectional view was carried out using ion milling system (IM-40000, Hitachi). Dual-beam focused ion beam (Helios 450HP, FER) was used for cross-section view of TEM images. To observe the volume expansion of the electrode at lithiated state after cycling, the cells are disassembled, and the electrodes are rinsed with dimethyl carbonate in a dry room. Specific surface area and mesopore diameter distribution were estimated with the Brunauer–Emmett–Teller (BET) method using the nitrogen adsorption–desorption analyzer (TriStar II, Micromeritics). Prior to measurement, the samples were degassed at 120 °C for 2 h. And porosity and macropore diameter distribution were determined by mercury-porosimetry (Autopore V9500, Micromeritics).

**Electrochemical characterization**. Working electrode was prepared by mixing slurry composed of the active material (G, GS, MGS), the conductive agent (Super P, TIMCAL), sodium carboxymethylcellulose (CMC) and styrene butadiene rubber (SBR) at mass ratio of 96:1:1.5:1.5 and then the homogeneously blended slurry was cast onto the Cu current collector up to 6.9 mg cm$^{-2}$. In the case of graphite, the loading level was 10.1 mg cm$^{-2}$ for same areal capacity. All electrodes were dried at 80 °C for 0.5 h and then calendared for 1.6 g cm$^{-3}$ of electrode density with the electrode thickness of 43 μm of the MGS and GS electrode and 63 μm of the G electrode excepting the Cu current collector. In sequence, the electrode was finally vacuum-dried at 110 °C for 8 h. The cathode electrode was made by casting slurry on a Al current collector with conventional lithium cobalt oxide, carbon black, and polyvinylidene fluoride binder in a mass ratio of 96:2:2. The mass loading level of the cathode was 20 mg cm$^{-2}$ and then pressed until the density of the electrode became 3.6 g cm$^{-3}$ with the electrode thickness of 55 μm excepting for the Al current collector. N-methyl-2-pyrrolidone (NMP) was used as the solvent. CR2032 (half-cell) and pouch (full-cell) type cells were assembled in the dry room using these working electrodes. The electrolyte was 1.3 M LiPF$_6$ in mixture of ethylene carbonate/ethyl methyl carbonate/diethyl carbonate (3/5/2, by volume) with 10% of fluoroethylene carbonate, 0.2% of lithium tetrafluoroborate, 0.5% of vinylene carbonate, 3% of succinonitrile, and 1% of propane sultone (Panax Starlyte) and microporous polyethylene was used as a separator with a thickness of 20 μm. As a counter electrode, pure Li metal foil (1 mm) was used for half-cell, and LCO was utilized for full-cell. Electrochemical tests of the half-cell were carried out in a voltage range of 0.005–1.5 V at 0.35 mA cm$^{-2}$ (0.1 C) for the first cycle, and between 0.005 and 1.0 V at 1.75 mA cm$^{-2}$ (0.5 C) for the rest of the cycles. Electrochemical performances of the full-cell, designed with an N/P ratio of 1.1, were evaluated in the voltage range between 2.5 and 4.35 V at 0.1 C for the first cycle, and between 2.7 and 4.35 V at a discharge rate of 1 C and charge rate of 0.5 C for the rest of the cycles. The electrolytes and separator in the full-cell were the same as those in the half-cell above. All the cell tests were done using a battery cycler (TOSCAT-3100, TOYO SYSTEM) at 25 °C.

**Dilatometry**. Thickness change of pouch (full-cell) type cell during 100 cycles was measured using an electrochemical dilatometer (Mitutoyo). All electrochemical conditions were the same as those in the full-cell above.

**Numerical simulations**. For finite element simulations, we used the commercial software ABAQUS/Standard 6.14 (Dassault System). Thermal-mechanical coupled model that is equivalent to the lithium-diffusion model was utilized to describe the volume expansion and contraction of GS and MGS particles during lithiation and delithiation processes. Two-phase lithiation model of Si[34] and linear elastic materials with concentration-dependent mechanical properties which had been measured by experiments[35,36] were adopted in all simulations. To reduce computational cost, we employed circular-shaped models of which dimensions were smaller than the GS and MGS particles in the experiment. Radiuses of the particles, macropores, and mesopores were 1100, 500, and 20 nm, respectively. Thicknesses of the Si-layer on macropores in the GS and MGS particles were 20.0 and 36.8 nm, respectively, so that the total weight percent (wt%) of Si for both GS and MGS particles was the same. In static fracture simulations, we calculated fracture energies (i.e., J-integrals) on five independent pathways encircling a pre-existing nanometer-sized crack (5 nm) on the graphite side at the interface of the mesopores and compared with critical fracture energies (i.e., fracture toughness) of the corresponding materials. We note that we intentionally set the models stress-free at the beginning of lithiation and delithiation processes during the fracture simulations, and thus the residual stress field resulting from the lithiation process was not considered when we started the delithiation process. We did not conduct crack propagation simulations.

## Data availability

The authors declare that the data supporting the findings of this study are available within the article and its Supplementary Information Files. All other relevant data supporting the findings of this study are available on request.

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

## Acknowledgements

This work was supported by the Korea Institute of Energy Technology Evaluation and Planning (KETEP) and the Ministry of Trade, Industry & Energy (MOTIE) of the Republic of Korea (No. 20172410100140). Also, financial support from the 2019 Research Fund of UNIST is greatly acknowledged.

## Author contributions

J.M., J.S., and M.K. conceived and designed the experiments. J.M. and J.S. prepared the samples and carried out the main experiments including construction of models for G, GS, and MGS. J.H. performed structure and fracture analyses using FEM. S.C. conducted dilatometry. N.K. participated in electrochemical measurements. S.-H.C. and G.N. assisted with sample preparation. Y.S. provided experimental insights into the electrochemical test. J.M., J.S., J.H., S.Y.K., M.K., and J.C. co-wrote the paper. J.M., J.S., J.H., S.Y.K., M.K., and J.C. discussed the results and revised or commented the manuscript.

## Additional information

**Competing interests:** The authors declare no competing interests.

