## [Peer Review File · Nature Communications]

Reviewers' comments:

Reviewer #1 (Remarks to the Author):

This manuscript presents a new way to synthesize silicon/graphite composite anode to achieve high volumetric capacity in full cells. Particularly, the authors claimed that a preliminary deposition of carbon to fill the mesopores within the graphite before silicon deposition within the macropores of the graphite is crucial to obtain good cyclic stability. Because of the structure characteristic of the graphite anode, it is able to achieve high volumetric capacity with the silicon/graphite composite anode. The idea of preparation of silicon/graphite composite anode by CVD deposition is not very new, which has been addressed by several groups. However, the role of mesopores and macropores within the graphite on the structure and performance of the silicon/graphite composite anode has not been addressed so far. Besides, the insight about the relationship between the carbon/silicon deposition and volumetric capacity is new. Furthermore, conceptual ideas about the volumetric capacity and cyclic stability behavior of the silicon/graphite composite anode have been addressed in depth by the authors with mathematical and mechanical simulation tools. In general, the authors have made a few novel claims regarding the high performance silicon/graphite composite anode, which can help the community gain new thinking and insight about a few critical issues related to the silicon based anode. The quality of the manuscript is good and worth publishing in Nature Communications. However, the manuscript still has a few important questions to address before it can be considered for acceptance of publication in Nature Communications. Details are as following:

1. As pointed out by the authors, the carbon filling in the mesopores and deposition of silicon in the macropores of the graphite are crucial for cyclic stability improvement and high volumetric capacity. The relationship between the mesopores and macropores including relative composition, extents of carbon filling and silicon deposition are actually very important. It seems that a delicate control over these parameters is critical for the volumetric capacity. The author claimed in the manuscript as "Maximized Volumetric Capacity via Pore-coordinated Design for Large-volume-change Li-ion Battery Anodes", but it is not clear whether the experimental protocols explored in this manuscript can really maximize the volumetric capacity based on the silicon/graphite anode. Further systematic experiments seem necessary to prove that the volumetric capacity is maximized with the experimental protocol applied in this manuscript.

2. The authors applied preliminary carbon deposition to fill the mesopores and on the surface of the macropores as well. The detrimental effect of the mesopores acting as a host for silicon deposition is inhibited. It is critical for the concept of this manuscript. However, the deposited silicon may also contain mesopores, which is not addressed by the authors. It may not be reflected by the porosity measurement because the mass composition of the deposited carbon may be much lower compared to the graphite matrix.

3. Considering that the size of the graphite is in the micrometer scale. It is not clear whether the carbon deposition and silicon deposition homogeneously throughout the whole graphite particles, or some gradient existing within the graphite particles. The spatial distribution behavior of the carbon and silicon deposition may influence the final performance of the silicon/graphite composite anode.

4. The work about the deposition of silicon on graphite was explored previously. The merit of this manuscript is to have some unique insights about this method. It would be better for the authors to cite those references, even though those papers did not address the role of mesopores and macropores of the graphite on the silicon deposition.

5. The authors emphasize the importance of the volumetric capacity, which is important for automotive applications. However, the authors demonstrated the application of the high volumetric capacity Si/graphite anode based on LiCoO₂ cathode, which is seldom used in the automotive applications. It would be more straightforward and strong support if the authors could provide data with typical cathode for automotive applications.

6. It would be helpful if the authors could provide practical relevant performance requirement

regarding the volumetric capacity to help readers identify what the level of the current work has achieved.

Reviewer #2 (Remarks to the Author):

In the manuscript titled "Towards Maximized Volumetric Capacity via Pore-coordinated Design for Large-volume-change Li-ion Battery Anodes", the rational design of a macropore-coordinated graphite-Si composite greatly contemplated the electrode swelling upon cycling, exhibiting impressive volumetric capacity in a full-cell configuration. The authors presented a comprehensive understanding of the mechanism of the novel structure, combining simulation and mathematical calculation with experimental data. The feasible fabrication process and improved volumetric capacity and energy density, with much reduced volume expansion, promises a possibility in commercializing silicon as the next-generation anode for electric vehicles in the near future.

The manuscript is suggested for publication after addressing the following issues.

1. One of the authors previous works, cited as Ref [23], studied the graphite-silicon composite applying the same porous structure to accommodate the volume change during cycling. This work leans one step further to control the filling of Si into mesopores or macropores. A model has been used to calculate the thickness of Si-layers as a function of the radius of macropore. In the case of MGS, only one set of Si 'loading' was examined. Within the thickness limit in macropores, is it possible that different thickness of Si would lead to even smaller volume expansion upon lithiation? How will this trade off with the volumetric capacity? Please provide additional set of evaluation results if possible.
2. Following up the first question, the authors use carbon-blocking instead of Si-blocking when comparing MGS and GS. To what extent will the carbon-blocking prevent the later Si deposition occupying the undesired mesopore sites? In other words, how to determine how much carbon is grown sufficiently in this poor-coordinated structure? How to correlate the amount of carbon grown (the size of mesopores blocked) with the thickness of the Si layer grown in succession?
3. The stress produced during lithiation was simulated in the last paragraph on page 7. Different mechanisms were proposed for the mesopores and macropores. Please supplement TEM results as supports, either in-situ or ex-situ results.
4. In Figure 4, the SEM images comparing GS and MGS are of different scale. It would be better if images with same scale are used.
5. In the section discussing "Morphological changes and SEI formation" on Page 8, the authors mentioned in line 178: "Previous studies have reported Si nanoparticle critical sizes of approximately 150 and 300 nm, and Si fracturing can be avoided under these sizes^{30,31}. Although < 50 nm is far smaller than the Si critical size, isolated Si-blocking seems to cause different morphological changes upon cycling." The question is raised and the explanation seems too simplified. In the case of GS, Si fracturing is proposed in the manuscript as one of the degradation mechanisms. Please provide further discussion on why the Si-filling in the mesopores < 50 nm would still lead to Si fracturing.
6. In Figure 4 f and j, TEM (STEM) of electrodes after cycling were compared as evidence of fractures and SEI formation. First, please highlight the areas in Figure 4 e and i where f and j are taken if they are the zoom-in of selected areas. If not, please label what is the dark area in the mapping, whether they are the cavity in macropore, graphite cracking or Si fracture. In addition, the mapping of F and O

in these two sets could not prove the "SEI formation theory". In both cases, F and O are distributed everywhere except partial of the cavities. The density of F and O does not show much difference comparing Figure 4 f and j. Please provide other evidence showing the formation of SEI in both cases.

Reviewers' comments:

Reviewer #1 (Remarks to the Author):

This manuscript presents a new way to synthesize silicon/graphite composite anode to achieve high volumetric capacity in full cells. Particularly, the authors claimed that a preliminary deposition of carbon to fill the mesopores within the graphite before silicon deposition within the macropores of the graphite is crucial to obtain good cyclic stability. Because of the structure characteristic of the graphite anode, it is able to achieve high volumetric capacity with the silicon/graphite composite anode. The idea of preparation of silicon/graphite composite anode by CVD deposition is not very new, which has been addressed by several groups. However, the role of mesopores and macropores within the graphite on the structure and performance of the silicon/graphite composite anode has not been addressed so far. Besides, the insight about the relationship between the carbon/silicon deposition and volumetric capacity is new. Furthermore, conceptual ideas about the volumetric capacity and cyclic stability behavior of the silicon/graphite composite anode have been addressed in depth by the authors with mathematical and mechanical simulation tools. In general, the authors have made a few novel claims regarding the high performance silicon/graphite composite anode, which can help the community gain new thinking and insight about a few critical issues related to the silicon based anode. The quality of the manuscript is good and worth publishing in Nature Communications. However, the manuscript still has a few important questions to address before it can be considered for acceptance of publication in Nature Communications. Details are as following:

Authors' response:

We deeply appreciate the referee's insightful review of our manuscript. The referee's encouraging suggestions have been carefully considered and thoroughly addressed as listed below.

1. As pointed out by the authors, the carbon filling in the mesopores and deposition of silicon in the macropores of the graphite are crucial for cyclic stability improvement and high volumetric capacity. The relationship between the mesopores and macropores including relative composition, extents of carbon filling and silicon deposition are actually very important. It seems that a delicate control over these parameters is critical for the volumetric capacity. The author claimed in the manuscript as "Maximized Volumetric Capacity via Pore-coordinated Design for Large-volume-change Li-ion Battery Anodes", but it is not clear whether the experimental protocols explored in this manuscript can really maximize the volumetric capacity based on the silicon/graphite anode. Further systematic experiments seem necessary to prove that the volumetric capacity is maximized with the experimental protocol applied in this manuscript.

Authors' response:

We are grateful for the reviewer's constructive suggestion and agree with the opinion of the reviewer that experimental protocol for finding out maximized volumetric capacity is needed. To investigate the optimum values of the maximized volumetric capacity, we first identified the ethylene flow time for the carbon filling in MG through controlling the mesopore size distribution. Several-typed MGSs synthesized with different ethylene flow time for 1.5 h (1.5h-MGS), 3 h (3h-MGS), 4.5 h (4.5h-MGS) and 6 h (6h-MGS) were compared as shown in **Figure R1 (Supplementary Fig. 11)**, when all samples exhibit the similar

gravimetric capacities. Among the various MGSs, 3h-MGS demonstrates the best cycle retention and lowest electrode swelling ratio during full-cell cycling. On this account, we confirmed the ethylene flow time for 3 h is the optimal condition for the maximized volumetric capacity as presented in the manuscript (MGS). The reason why the excessive carbon filling samples of 4.5h-MGS and 6h-MGS demonstrate capacity degradation is the poreless (poreless means that the samples scarcely contain pores) characteristics. Such phenomenon corresponds with the case of the aGS which scarcely contain pores (Supplementary Fig. 7). To be specific, when the Si is coating on the graphite without pores, it leads to the thick silicon layer with the thickness of ~ 90 nm and coated graphite does not offer enough space for accommodating the Si volume expansion. On the contrary, the 1.5h-MGS still contains lots of mesopore because of its insufficient ethylene flow time, leading to the generation of Si filling during Si coating process, which results in the poor cycling stability and high electrode swelling ratio (Figure R2 as Supplementary Fig. 12). As a result, now that minimum and maximum amount of carbon blocking could give rise to the low volumetric capacity considering the cycling retention and electrode swelling ratio, we finally conclude that 3h-MGS is the best demonstrating the maximized volumetric capacity. Overall, all related descriptions have been added in Supplementary Note 4.

Figure R1. Comparison of several-typed MGSs synthesized with different ethylene flow time for 1.5 h (1.5h-MGS), 3 h (3h-MGS), 4.5 h (4.5h-MGS) and 6 h (6h-MGS). (a) Mesopore size distribution determined via BJH method. (b) Voltage profiles for the first cycles at 0.1 C in half-cell and (c) cycling performances at 1 C in full-cell. (d) Thickness changes after the first lithiation of each sample.

Figure R2. (a) Voltage profile for the first cycle and (b) cycling performance of aGS at 0.5 C in half-cell. Cross-sectional views of aGS electrode and surface SEM images (c, e) before cycling and (d, f) after 100 cycles, respectively.

2. The authors applied preliminary carbon deposition to fill the mesopores and on the surface of the macropores as well. The detrimental effect of the mesopores acting as a host for silicon deposition is inhibited. It is critical for the concept of this manuscript. However, the deposited silicon may also contain mesopores, which is not addressed by the authors. It may not be reflected by the porosity measurement because the mass composition of the deposited carbon may be much lower compared to the graphite matrix.

Authors' response:

Thank you very much for the reviewer's encouraging comment. We agree with the opinion that the deposited silicon may have some mesopore. Fortunately, as shown in Fig. 2g and i, the pore volume in mesopore range and specific surface area are reduced after Si deposition (to GS and MGS from G and MG).

These results suggest that the total amount of mesopores can be reduced by silicon deposition. Also, as shown in **Figure R3**, MGS with high silicon deposition (15 wt% Si) has a lower specific surface area ($1.5\text{ m}^2\text{ g}^{-1}$) and a higher ICE (93.7%) than those of MGS (6.3 wt% Si, $2.1\text{ m}^2\text{ g}^{-1}$ and 93%). Also, in the TEM cross-section image in Fig. 2f, it was confirmed that Si is coated without any pores.

Figure R3. (a) Voltage profile for the first cycle, (b) specific surface area and initial coulombic efficiency of MGS with high silicon deposition (15 wt% Si) compared to those of MGS (6.3 wt% Si).

3. Considering that the size of the graphite is in the micrometer scale. It is not clear whether the carbon deposition and silicon deposition homogeneously throughout the whole graphite particles, or some gradient existing within the graphite particles. The spatial distribution behavior of the carbon and silicon deposition may influence the final performance of the silicon/graphite composite anode.

Authors' response:

We agree with the reviewer's opinion that achieving the homogeneity of deposition on the graphite is a crucial issue. To minimize such a homogeneity problem, we developed a customized rotatable CVD furnace presented in **Figure R4** as **Supplementary Fig. 13** with related descriptions (Redacted). **Figure R4(c, e) and (b, d)** show high nonuniformity of silicon deposition in the non-rotation mode and superior uniformity of deposition in the rotation mode, respectively. In other words, we improved the homogeneity of the deposition through the rotation function during the process.

Figure R4. (a) The rotational driving system of customized rotatable CVD furnace. SEM images of MGS synthesized through (b, d) the rotation mode and (c, e) the non-rotation.

4. The work about the deposition of silicon on graphite was explored previously. The merit of this manuscript is to have some unique insights about this method. It would be better for the authors to cite those references, even though those papers did not address the role of mesopores and macropores of the graphite on the silicon deposition.

Authors' response:

We greatly appreciate the referee's helpful suggestion. The following 7 references were added to the

manuscript as advised.

25. Holzzapfel M, Buqa H, Krumeich F, Novák P, Petrat F-M, Veit C. Chemical Vapor Deposited Silicon/Graphite Compound Material as Negative Electrode for Lithium-Ion Batteries. *Electrochim Solid St* 8, A516-A520 (2005).
26. Zhang Y, et al. Composite anode material of silicon/graphite/carbon nanotubes for Li-ion batteries. *Electrochim Acta* 51, 4994-5000 (2006).
27. Lee J-H, Kim W-J, Kim J-Y, Lim S-H, Lee S-M. Spherical silicon/graphite/carbon composites as anode material for lithium-ion batteries. *J Power Sources* 176, 353-358 (2008).
28. Martin C, Alias M, Christien F, Crosnier O, Bélanger D, Brousse T. Graphite-Grafted Silicon Nanocomposite as a Negative Electrode for Lithium-Ion Batteries. *Adv Mater* 21, 4735-4741 (2009).
29. Yoon YS, Jee SH, Lee SH, Nam SC. Nano Si-coated graphite composite anode synthesized by semi-mass production ball milling for lithium secondary batteries. *Surface and Coatings Technology* 206, 553-558 (2011).
30. Gan L, et al. A facile synthesis of graphite/silicon/graphene spherical composite anode for lithium-ion batteries. *Electrochim Acta* 104, 117-123 (2013).
31. Xiao C, He P, Ren J, Yue M, Huang Y, He X. Walnut-structure Si-G/C materials with high coulombic efficiency for long-life lithium ion batteries. *RSC Advances* 8, 27580-27586 (2018).

5. The authors emphasize the importance of the volumetric capacity, which is important for automotive applications. However, the authors demonstrated the application of the high volumetric capacity Si/graphite anode based on LiCoO₂ cathode, which is seldom used in the automotive applications. It would be more straightforward and strong support if the authors could provide data with typical cathode for automotive applications.

Authors' response:

We apologize for the confusion caused by the full-cell results based LiCoO₂ (LCO) cathode that is seldom used in automotive applications. The reason why we have focused on the LCO cathode for the full-cell configuration was that our reference anode (commercial graphite) has often been paired with the LCO in industrial application. According to the reviewer's insight comments, we suggest the results of the pouch full-cell based on MGS and LiNi_{0.6}Co_{0.2}Mn_{0.2}O₂ which is widely used in automotive applications as shown in **Figure R5 (Supplementary Figure 14)**.

Figure R5. Voltage profiles and cycling performances of $\text{LiNi}_{0.6}\text{Co}_{0.2}\text{Mn}_{0.2}\text{O}_2$, which is widely used in automotive applications, (a, b) half-cell and (c, d) full-cell paired with the MGS, respectively.

6. It would be helpful if the authors could provide practical relevant performance requirement regarding the volumetric capacity to help readers identify what the level of the current work has achieved.

Authors' response:

We thank the reviewer for this constructive suggestion. Since the spherical graphite used in the manuscript is already commercially used, we believe that the volumetric capacity of this graphite is criteria for practical relevant performance. In the case of Fig. 5, the comparison of volumetric capacity was made based on only the negative electrode and not including copper foil. To provide further practical relevant performance, **Figure R6 (Supplementary Figure 15)** presents the comparison of volumetric capacity including both electrodes, current collectors and membrane at the industrial level. In addition, we found out incorrect calculated values in manuscript and supporting information. Thus, **we have revised these values after careful consideration.**

Figure R6. Schematic illustration including thickness information of pouch full-cell at the industrial level comprising graphite and MGS (a) before cycling and (b) after the first lithiation. (c) Total thickness and volumetric capacity of MGS and graphite full-cell, considering the thickness of both electrodes, current collectors and membrane.

Reviewer #2 (Remarks to the Author):

In the manuscript titled “Towards Maximized Volumetric Capacity via Pore-coordinated Design for Large-volume-change Li-ion Battery Anodes”, the rational design of a macropore-coordinated graphite-Si composite greatly contemplated the electrode swelling upon cycling, exhibiting impressive volumetric capacity in a full-cell configuration. The authors presented a comprehensive understanding of the mechanism of the novel structure, combining simulation and mathematical calculation with experimental data. The feasible fabrication process and improved volumetric capacity and energy density, with much reduced volume expansion, promises a possibility in commercializing silicon as the next-generation anode for electric vehicles in the near future.

Authors' response:

Thank you very much for your careful and insightful review of our manuscript. You bring up several important points that we hope we can adequately address here.

1. One of the authors previous works, cited as Ref [23], studied the graphite-silicon composite applying the same porous structure to accommodate the volume change during cycling. This work leans one step further to control the filling of Si into mesopores or macropores. A model has been used to calculate the thickness of Si-layers as a function of the radius of macropore. In the case of MGS, only one set of Si 'loading' was examined. Within the thickness limit in macropores, is it possible that different thickness of Si would lead to even smaller volume expansion upon lithiation? How will this trade off with the volumetric capacity? Please provide additional set of evaluation results if possible.

Authors' response:

We gratefully appreciate the reviewer for this constructive comment. We would like to highlight that the Si 'loading' and 'thickness' are closely connected with the gravimetric capacity. Within the thickness limit in macropores, low amount of Si (same meaning the decrease of the Si thickness) brings about the low gravimetric capacity, leading to the low volumetric capacity. For these reasons, we calculated the required macropore radius for the maximum Si layer coating ratio (Supplementary Fig. 1 and Note. 2). In the case of excessive Si loading and thickness, electrode swelling ratio of MGS (15 wt% Si) with high silicon deposition is shown in **Figure R7**. Due to the limit of the amount of macropore volume in the graphite, the electrode swelling ratio increases sharply as the amount of silicon becomes excessively large. Also, cyclability of graphite-silicon composite rapidly deteriorates. These results imply that volumetric capacity fading can occur during cycling. Therefore, when considering the trade-off relation between Si loading (thickness) and volumetric capacity, currently suggested the volumetric capacity of MGS demonstrated the maximum level. Redacted

Redacted

Figure R7. (a) Thickness change after the first lithiation and (b) cycling performance of MGS with high silicon deposition (15 wt% Si) compared to those of MGS (6.3 wt% Si). Redacted

2. Following up the first question, the authors use carbon-blocking instead of Si-blocking when comparing MGS and GS. To what extent will the carbon-blocking prevent the later Si deposition occupying the undesired mesopore sites? In other words, how to determine how much carbon is grown sufficiently in this poor-coordinated structure? How to correlate the amount of carbon grown (the size of mesopores blocked) with the thickness of the Si layer grown in succession?

Authors' response:

Thank you very much for the reviewer's valuable suggestions. We agree that the determination in the level of sufficient carbon-blocking is an important issue. To investigate the optimum values of carbon fillings, we first identified the ethylene flow time for the carbon filling in MG through controlling the mesopore size distribution. Several-typed MGSs synthesized with different ethylene flow time for 1.5 h (1.5h-MGS), 3 h (3h-MGS), 4.5 h (4.5h-MGS) and 6 h (6h-MGS) were compared as shown in **Figure R1 (Supplementary Fig. 11)** when all samples exhibit the similar gravimetric capacities. Among the various MGSs, 3h-MGS demonstrates the best cycle retention and lowest electrode swelling ratio during full-cell cycling. On this account, we confirmed the ethylene flow time for 3 h is the optimal condition for maximized volumetric capacity as presented in the manuscript (MGS). And, the reason why the excessive carbon filling samples of 4.5h-MGS and 6h-MGS demonstrate capacity degradation is the poreless (poreless means that the samples scarcely contain pores) characteristics. Such phenomenon corresponds with the case of the aGS which scarcely contain pores (Supplementary Fig. 7). To be specific, when the Si is coating on the graphite without pores, it leads to the thick silicon layer with the thickness of ~90 nm and coated graphite does not offer enough space for accommodating the Si volume expansion. The comparison of GS, MGS and aGS

reveals that the Si thickness becomes thicker as the pore volumes are decreasing (Figure R8 as Supplementary Fig. 16). On the contrary, the 1.5h-MGS still contains lots of mesopore because of its insufficient ethylene flow time, leading to the generation of Si filling during Si coating process, which results in the poor cycling stability and high electrode swelling ratio (Figure R2 as Supplementary Fig. 12). As a result, now that minimum and maximum amount of carbon blocking could give rise to the low volumetric capacity considering the cycling retention and electrode swelling ratio, we finally conclude that 3h-MGS is the best demonstrating the maximized volumetric capacity. Overall, all related descriptions have been added in Supplementary Note 4.

Figure R8. Porosity and silicon thickness of GS, MGS and aGS, respectively.

3. The stress produced during lithiation was simulated in the last paragraph on page 7. Different mechanisms were proposed for the mesopores and macropores. Please supplement TEM results as supports, either in-situ or ex-situ results.

Authors' response:

We thank the reviewer for this useful suggestion. Ex-situ TEM images after the initial expansion of silicon in mesopore and macropore are attached to Figure R9 (Supplementary Fig. 17). *In situ* TEM results for silicon expansion in mesopore and macropore can also be found in Ref [23]. Also, this information has been added in the manuscript.

Figure R9. TEM images after the initial expansion of silicon in (a) mesopore and (b) macropore.

4. In Figure 4, the SEM images comparing GS and MGS are of different scale. It would be better if images with same scale are used.

Authors' response:

Thank you very much for the reviewer for this detailed comment. We have revised according to following the reviewer's suggestion and added previous **Fig. 4g and i** to **Figure R10 (Supplementary Fig. 18)**.

Figure R10. Magnified SEM images of (a) Fig. 4g and (b) i.

5. In the section discussing “Morphological changes and SEI formation” on Page 8, the authors mentioned in line 178: “Previous studies have reported Si nanoparticle critical sizes of approximately 150 and 300 nm, and Si fracturing can be avoided under these sizes^{30,31}. Although < 50 nm is far smaller than the Si critical size, isolated Si-blocking seems to cause different morphological changes upon cycling.” The question is raised and the explanation seems too simplified. In the case of GS, Si fracturing is proposed in the manuscript as one of the degradation mechanisms. Please provide further discussion on why the Si-filing in the mesopores < 50 nm would still lead to Si fracturing.

Authors' response:

We are grateful for the reviewer's valuable advice. We would like to demonstrate the reason why the Si filling in the mesopore < 50 nm leads to Si fracturing from the viewpoint of mathematical calculation and morphological aspect.

- (1) Mathematical calculation of stress intensification to the Si layer on the mesopore and macropore during lithiation.

Through FEM results in Fig. 4k-m, we describe our interpretation about the reason for morphological deformation and fracture of Si below 50 nm as follows: "Noting that the critical fracture energies for Si and G crack formation are reported to be 9 and 17–69 J m⁻², respectively, we examined the possible growth of a crack applied to the Si and G adjacent to the Si-blocking. The calculated Si and G fracture energies with a pre-existing 5 nm-sized crack in GS during lithiation were 10.44 and 71.04 J m⁻², respectively (6.88 and 3.80 J m⁻², respectively, in MGS); thus, results exceeding the critical values indicate crack growth during the lithiation/delithiation process". In other words, considering the higher calculated fracture energies (10.44 J m⁻²) than the critical fracture energies for Si (9 J m⁻²), the silicon filling in the mesopore can be deformed and fractured. We have revised our manuscript by following the reviewer's suggestions.

- (2) Morphological changes of Si layer on the mesopore and macropore during lithiation.

When the Si-blocking expands towards the G upon lithiation, it applies extremely high tensile stress on the G. The stresses allow the Si-blocking to attach strongly to the bottom and upper side of the G during lithiation. The adhesive Si maintains this contact upon contraction during delithiation, causing Si fracturing.

6. In Figure 4 f and j, TEM (STEM) of electrodes after cycling were compared as evidence of fractures and SEI formation. First, please highlight the areas in Figure 4 e and i where f and j are taken if they are the zoom-in of selected areas. If not, please label what is the dark area in the mapping, whether they are the cavity in macropore, graphite cracking or Si fracture. In addition, the mapping of F and O in these two sets could not prove the "SEI formation theory". In both cases, F and O are distributed everywhere except partial of the cavities. The density of F and O does not show much difference comparing Figure 4 f and j. Please provide other evidence showing the formation of SEI in both cases.

Authors' response:

Thank you very much for the reviewer for this insightful comment. First, we would like to inform the reviewer that SEM (Fig.4e and i) and TEM data (Fig. 4f and j) are not taken from the same area. When the milled sample (Fig. 4e and i) is etched for TEM sampling (Fig. 4f and j), the identifiable surface and pore is constantly changed. For this reason, we ask for your understanding that SEM and TEM data could not be taken from the same region. As following the reviewer's suggestions, we label dark area Fig. 4f and j. We also agree with the reviewer's opinion that there may be a debate on the "SEI formation theory". Therefore, we would like to suggest that continuous SEI layer can occur at the newly exposed surface by G cracks and Si-blocking fracturing. Since a cavity caused by Si-blocking fracturing appeared in Fig. 4f, we also insert the cavity at the continuous SEI-layer in Fig 4a. Considering these points, we have revised the manuscript. In addition, we found out incorrect calculated values in manuscript and supporting information. Thus, we have revised these values after careful consideration.

REVIEWERS' COMMENTS:

Reviewer #1 (Remarks to the Author):

All of the points raised by the reviewer have been addressed by the authors. The quality of the manuscript is improved, which enables it suitable for publishing in Nature Communications now.

Reviewer #2 (Remarks to the Author):

The revised manuscript has sufficiently addressed the questions raised by the reviewers. The mechanisms on how the mesopores and macropores could affect the performance were have been revealed. The authors carried out comprehensive and detailed work on the relation between carbon/silicon deposition and volumetric capacity. The study could be inspiring to the community of researchers who focus on silicon anodes. The manuscript is recommended for publication on Nature Communication.

Reviewers' comments:

Reviewer #1 (Remarks to the Author):

All of the points raised by the reviewer have been addressed by the authors. The quality of the manuscript is improved, which enables it suitable for publishing in Nature Communications now.

Authors' response:

We appreciate the reviewer spending the precious time and efforts on reviewing our manuscript.

Reviewer #2 (Remarks to the Author):

The revised manuscript has sufficiently addressed the questions raised by the reviewers. The mechanisms on how the mesopores and macropores could affect the performance were have been revealed. The authors carried out comprehensive and detailed work on the relation between carbon/silicon deposition and volumetric capacity. The study could be inspiring to the community of researchers who focus on silicon anodes. The manuscript is recommended for publication on Nature Communication.

Authors' response:

We would like to thank the reviewer's positive consideration for publication in Nature Communications.